# External Validation of a Prognostic Score for Survival in Lung Carcinoids

**DOI:** 10.3390/cancers14112601

**Published:** 2022-05-25

**Authors:** Marco Chiappetta, Diomira Tabacco, Carolina Sassorossi, Isabella Sperduti, Giacomo Cusumano, Alberto Terminella, Ludovic Fournel, Marco Alifano, Francesco Guerrera, Pier Luigi Filosso, Samanta Nicosia, Filippo Gallina, Francesco Facciolo, Stefano Margaritora, Filippo Lococo

**Affiliations:** 1Thoracic Surgery, Università Cattolica del Sacro Cuore, 00168 Rome, Italy; marco.chiappetta@policlinicogemelli.it (M.C.); sassorossi.caro@gmail.com (C.S.); stefano.margaritora@policlinicogemelli.it (S.M.); filippo.lococo@policlinicogemelli.it (F.L.); 2Fondazione Policlinico Universitario A. Gemelli IRCCS, LARGO A. Gemelli 8, 00100 Rome, Italy; 3Biostatistics, Regina Elena National Cancer Institute, IRCCS, 00128 Rome, Italy; isabella.sperduti@ifo.it; 4Thoracic Surgery, Policlinico-San Marco Hospital, 95121 Catania, Italy; giacomare55@hotmail.com (G.C.); albertoterminella0@gmail.com (A.T.); 5Department of Thoracic Surgery, APHP Centre—Université de Paris, 75000 Paris, France; ludovic.fournel@aphp.fr (L.F.); marco.alifano@aphp.fr (M.A.); 6Department of Thoracic Surgery, University of Turin, San Giovanni Battista Hospital, 10126 Turin, Italy; fra.guerrera@gmail.com (F.G.); pierluigi.filosso@unito.it (P.L.F.); samanta.nicosia@unito.it (S.N.); 7Thoracic Surgery Unit, Regina Elena National Cancer Institute IRCCS-IFO, 00128 Rome, Italy; filippogallina92@gmail.com (F.G.); francesco.facciolo@ifo.it (F.F.)

**Keywords:** lung carcinoid, lymph nodes, upstaging, lymphadenectomy

## Abstract

**Simple Summary:**

Incidence of lung carcinoids is rapidly increasing, but the correct management of these patients is still debated. Although their clinical behaviour differs from Non-Small Cell Lung Cancer, the same staging system is used for these tumors, even if it presents limitations in prognosis prediction and overlapping curves especially regarding sub-stages. For these reasons, in recent years, ad hoc scores have been constructed aiming to better stratify prognosis and indicate appropriate treatment options. In particular, a score including the node ration as nodal factor was proposed, although external validation was not possible. The aim of this study is to validate this score, for the possibility of identifying a specific class of patients that may benefit from specific follow-up schedules or post-operative treatments.

**Abstract:**

Background: A prognostic score including T-dimension, age, histology and lymph node ratio was previously proposed in absence of an external validation dataset. The aim of the current study was to validate the proposed prognostic score using an independent dataset. Methods: Data of patients with lung carcinoids, who underwent surgical resection and lymphadenectomy in five institutions from 1 January 2005 to 31 December 2019, were retrospectively analyzed. Two risk groups were created based on the following data: age, histology, node ratio and pT for disease-free survival (DFS); age, sex, node ratio and pT for overall survival (OS). The previously proposed score was validated, identifying two groups of patients: a high risk (HRG) and low risk (LRG) group. Results: The final analysis was conducted on 283 patients. Regarding DFS, 230 (81.3%) patients were assigned to the LRG and 53 (18.7%) to the HRG. Considering OS, 268 (94.7%) were allocated in the LRG and 15 (5.3%) in the HRG. The 5-year DFS was 92.7% in the LRG vs. 67% in the HRG (*p* < 0.001) while the 5-year OS was 93.6% in the LRG vs. 86.2% in the HRG (*p* = 0.29) with clear curve separation. Conclusion: Our analysis confirmed the validity of the composite score for DFS in lung carcinoids. Regarding OS, statistical significance was not reached because of a low number of deaths and patients in the HRG.

## 1. Introduction

Lung carcinoids (LC) are pulmonary neoplasms with increased incidence and prevalence in the last decades, which might be related to an increased detection rate for thorax computed tomography execution in lung screening protocols [1,2]. In 2008, the TNM staging system for lung tumours was adopted for LC [3], even if LC present a different clinical and survival outcome compared to Non-Small Cell Lung Cancer, with better long-term survival rates compared to other lung tumours [4]. In recent years, with the adoption of the 8th edition of the TNM staging system [5], it emerged that this staging system presents important limitations when LC were considered; survival curves frequently overlapped or statistical significance between groups, especially considering stage subcategories, was absent [6,7]. 

On the other hand, other studies analysed in depth the clinical and histological characteristics of patients affected by LC, building composite prognostic scores with the aim of improving prognosis stratification in these patients. For instance, Filosso et al. [8] proposed an ad hoc prognostic model for typical carcinoids. Instead, Cattoni et al. [7] presented a staging revision of the TNM, based on the dimension and the histology of all lung neuroendocrine tumours (also including large cell and non-small cell lung cancer). In particular, Chiappetta et al. [9] proposed specific prognostic scores for disease-free (DFS) and overall survival (OS) in LC only, which aimed to identify other lymph node factors for prognosis predictors, including the node ratio in the score confirming the importance of this parameter in LC. 

All the cited articles [6,7,8,9] pointed out the importance of different parameters and prognostic factors other than the TNM in LC management, suggesting the importance of a global approach to LC patients which also considers clinical factors. Moreover, it seems that nodal involvement is strong prognosticator in LC [10,11,12], with a worse prognosis in patients with nodal involvement compared to N0 patients. It is also important to note that although the TNM classified N1 and N2 nodes, the prognosis in LC with nodal metastases is poor by the anatomical location [10,11,12], suggesting the possibility of a different nodal parameter than the anatomical classification for prognosis stratification in these patients. Finally, it is not clear if adjuvant therapy may provide a survival benefit in patients with nodal involvement, and this issue may be related to non-appropriate patient selection for post-operative treatments [13,14]. On the other hand, the use of appropriate scores may also permit identification of patients that may really benefit from adjuvant treatments for survival improvement. 

The previous proposed score by Chiappetta et al. [9] was the first to evaluate different nodal parameters in LC (node ratio) and permitted a good prognosis stratification including other tumour and patient characteristics, with promising results in that context. However, the study included only an internal validation, and an independent dataset for external validation was not available. The aim of this current study is to validate the previous prognostic score, using an independent dataset to test the validity of this score in patients affected by LC.

## 2. Materials and Methods

### 2.1. Patients

Data regarding patients who underwent surgery for LC in five different centres, between 1 January 2005 to 31 December 2019, were collected and retrospectively analysed. Inclusion criteria for patients were age > 18 years, absence of distant metastases or contralateral nodal disease, preoperative computed tomography with contrast, pathological diagnosis of LC, anatomical lung resection, any kind of lymph node assessment (sampling, lobe specific dissection or mediastinal radical node dissection). Exclusion criteria were non-anatomical resection, absence of lymph node assessment and incomplete follow-up data or a follow-up shorter than 24 months. 

The histological diagnosis and discrimination between typical carcinoids (TC) and atypical carcinoids (AC) was conducted using immunohistochemistry, presence of necrosis and mitotic activity [15]: TC was defined as highly organized carcinoid architectures with less than two mitoses/10 high-power fields (HPFs). AC was characterized by a greater mitotic activity (2–10/10 HPF) and focal or discrete necrosis.

Surgical treatment was indicated after multidisciplinary discussion, and preoperative management was similar among the different centres: A preoperative CT scan with contrast was always performed, while a 18FDG-PET or PET with somatostatin analogues (dotanoc-, dotatoc- or dotatate-PET) was performed if indicated, after it became available. Finally, pre-operative diagnosis was obtained through thoracic needle ago-biopsy in case of peripheral tumours, via bronchoscopy in case of central tumours when technically feasible. If preoperative diagnosis was not available, an intraoperative frozen section of the nodule was performed to rule out the presence of neoplasm and differentiate from non-neoplastic nodules. Minimally invasive mediastinal staging by EBUS or EUS was performed in selected cases to prove nodal involvement. 

Tumours were considered central if directly visible during bronchoscopy or included in the first third of the lung at CT imaging. They were considered peripheral if not visible during bronchoscopy and located out the first third of the lung at the CT scan, or judged resectable with a wedge resection. 

Surgery was performed via thoracotomy or VATS and consisted in anatomical resections (segmentectomy, lobectomy or bilobectomy, pneumonectomy) and sleeve resections were performed whenever possible instead of pneumonectomy. 

The IASLC nodal map was used for lymph node station identification [16] and lymphadenectomy was performed according to the ESTS guidelines as follows [17]:

Node sampling: One or more lymph nodes thought to be representative were removed, guided by preoperative or intraoperative findings. 

Lobe-specific systematic node dissection: Specific lymph node stations were excised depending on the lobar location of the primary tumour.

Radical nodal dissection: The entire mediastinal tissue, containing the lymph nodes, was dissected and systematically removed within anatomical landmarks. 

The surgeon decided the lymphadenectomy extent based on professional experience, tumor location, histology and dimensions. Pathological reports were reviewed counting the number of resected lymph nodes and metastatic nodes, for lymph node ratio (NR) calculation. 

Follow-up consisted of physical examination, blood analysis and radiological examination (computer tomography and eventually fluorine-18-fluorodeoxyglucose positron emission tomography/computed tomography 18F-FDG PET/CT]) or PET with somatostatin analogues (dotanoc-, dotatoc- or dotatate-PET) every 6 months for the first two years and then annually. Patients were classified free from disease when medical examination and follow-up tests were negative for suspected relapses or metastasis.

### 2.2. Previous Prognostic Score

The prognostic score previously proposed consisted of four variables for OS and DFS (Table 1) [9].

In detail, the previous score was generated using a multi-centric dataset of 223 patients affected by lung carcinoids who underwent surgery with lymph node assessment. The log-HR obtained from the Cox model was used to derive weighting factors of a continuous prognostic index designed to identify differential risk outcomes. Coefficient estimates were ‘normalized’ by dividing by the smallest and rounding the resulting ratios to the nearest value. Thus, a continuous score assigning an ‘individualized’ risk to patients was generated. The score was dichotomized according to maximally selected log-rank statistics. To address the multivariate model in terms of fit and to validate the results, a cross-validation technique evaluated the replication stability of the final Cox multivariate model, using a resampling procedure. In this way, age, sex, lymph node ratio and pT were included in the OS score, while age, histology, lymph node ratio and pT were included in the DFS score. Using these factors, and adding the different points according to the risk factors, it was possible to identify two categories of patients: a low risk group (LRG), with a score < 1.5 for DSF and a score ≤ 3.1 for OS, and a high risk group (HRG), with a score ≥ 1.5 for DFS and a score > 3.1 for OS [9].

### 2.3. Statistical Analysis

Descriptive statistics were used to describe patients’ characteristics. Normality of continuous variables was investigated with the Kolmogorov–Smirnov test. Normal continuous variables were expressed with mean and standard deviation (SD), whereas non-normal variables were expressed using the median (interquartile interval). Categorical variables were expressed using frequencies. OS and DFS were calculated with the Kaplan–Meier product-limit method. OS was calculated starting from the date of intervention to the date of death from any cause or the date of the last follow-up visit, DFS was calculated from time of surgery to time of local relapse, distant metastasis appearance or death. If a patient was living, survival was censored at the time of the last visit. The log-rank test was used to assess differences between subgroups. The hazard risk (HR) and the confidence limits were estimated using the Cox univariate model, adopting the most suitable prognostic category as the referent group. Significance was defined at *p* < 0.05. The SPSS (v. 21.0, SPSS Inc., Chicago, IL, USA) statistical program was used for all analyses. In the present project, there are no ethical problems or undeclared conflicts of interest.

## 3. Results

Data of 415 patients who underwent surgery for LC were collected, and the final analysis was conducted on 283 patients who met the inclusion criteria (Figure 1). 

Clinical and pathological characteristics are reported in Table 2. 

TC were the predominant histology, present in 84.5% of cases. Lobectomy was performed in 84.1% of patients, while a radical mediastinal lymph node dissection was performed in the majority of cases (76.6%). During the study period, recurrences occurred in 37 (13.1%) patients and 23 (8.1%) patients died. In detail, LC-related death due to tumour progression occurred in 20 patients. The median follow-up was 50 months (range 24–237). Regarding the other prognostic score variables, male patients numbered 116 (40.9%), age < 61 years was 135 (47.7%), T2–T3 tumours occurred in 56 (19.8%) patients, while a node ratio <10% was present in 251 (88.7%) patients.

### Prognostic Score Validation

According to the risk categories, the results placed 230 (81.3%) in the LRG and 53 (18.7%) in the HRG for DFS; 268 (94.7%) were in the LRG and 15 (15.3%) in the HRG for OS. Patients’ characteristics according to risk classes are reported in Table 3.

5-year DFS was 92.7% in the LRG vs. 63.0% in the HRG (HR 5.026; 95% CI 2.635–9.588, *p* < 0.001) (Figure 2), while 5-year OS was 93.6% vs. 86.2% (HR 1.904; 95% CI 0.562–6.447, *p* = 0.30) (Figure 3).

## 4. Discussion

We confirm by means of an external dataset the potential validity of the previous proposed prognostic scoring system for LC [9], for survival prediction in LC patients. The score for DFS is particularly effective, permitting a clear recurrence rate distinction between the two groups suggesting different postoperative management courses for each category. Histology, tumor dimension and NR were included in the DFS subgroups, which is reasonable when planning ad hoc surveillance schedules for the HRG with more frequent and close follow-up. Interestingly, current guidelines regarding LC management are predominantly focused on different treatment stages, while limited information is present regarding post-operative schedules and follow-up [18,19]. The ESMO guidelines for lung and thymic carcinoids [19] also suggest long-life follow-up for these patients, recommending schedules considering histology, T-dimension and nodal status, but leaving a sort of physician discretion regarding the schedules especially for patients with nodal involvement or atypical histology. In our study, the 5-year DFS was 92.7% in the LRG, suggesting that in these patients delayed follow-up for a limited number of years may be applicable from a safety perspective. On the other hand, we identified a high-risk group of patients that presented a 5YDFS of 63%, about the 30% lower than the LRG. The most important finding concerns these high risk patients, who should be the objective of a rigid follow-up especially during the first years after surgery. Indeed, in LRG patients an annual follow-up for the first five years may be enough, while in HRG follow-up may be scheduled every 6 months and maybe longer than five years, even if we need further longer studies to validate this hypothesis. 

Another fundamental point concerns adjuvant therapy in LC, a subject still under debate, and no definitive results are available considering its administration [13,14]. Wegner et al. [14] did not report any survival advantage in patients undergoing adjuvant therapy vs. observation in both stage I/II and stage III atypical carcinoids, including after propensity matching. It is important to note that patients reported a survival difference according to the lymph node ratio, but specific analysis in the different node ratio sub-groups was not performed. Moreover, adjuvant therapy is not actually indicated in case of nodal involvement [18,19], but the indications are to identify patients at high risk of recurrence after multidisciplinary discussion. A score that allows patients at high risk of recurrence to be identified may be a valuable tool for patient management and adjuvant therapy indications, and may be the base for further prognostic studies or retrospective analysis.

Composite scores are now considered for NSCLC patient management [20], and, in our opinion, the adoption of a score in LC that permits the synthesis of all considered variables, may be a useful and effective tool to help physicians in their patient management. Indeed, this score included all the variables considered in the ESMO guidelines, adding the age variable, which seems to be an insignificant prognostic factor in these tumours, as Filosso et al. underlined [8].

Furthermore, the ESMO guidelines considered the nodal involvement independently from occurrence in hilar or mediastinal lymph nodes. We believe that this is an important point, and it is one of the reasons to use in the score a derivate parameter such as the lymph node ratio, which may reflect two crucial aspects in LC management: the extent of lymphadenectomy and metastatic spread.

Although the TNM staging system is adopted in LC, available literature seems to demonstrate that the presence of nodal involvement itself may be enough for prognosis stratification in these patients, independently from the nodal zone (N1 or N2) [10,11,12]. On the other hand, no available guidelines regarding lymph node assessment are available in literature, and only few studies have analysed the potential role of lymphadenectomy in these patients. Cattoni et al. did not report significant survival differences comparing sampling vs. radical lymphadenectomy, but patient and lymph node numbers associated with the two techniques were missing in the paper; therefore, it is difficult to know if they should be interpreted in terms of harvested nodes or node stations [7]. Contrariwise, other studies reported the prognostic role of lymphadenectomy, demonstrating a worse prognosis in patients with limited or absent nodal assessment, which suggests the importance of lymph node assessment in LC [21,22]. For these reasons, the adoption of a derivate parameter such as the node ratio may be extremely useful in LC, especially to gain information on the nodal status, but also on nodal spreading and extent of lymphadenectomy.

A recent study, presented at the 34th EACTS annual meeting [23], reported a discrete rate of mediastinal and nodal upstaging in LC, especially in atypical and central tumours, with a remarkably increased rate in case of radical mediastinal dissection. Moreover, on average one positive node was harvested, which suggests that the extent of lymphadenectomy is a fundamental part of LC staging and post-operative management.

For these reasons, the presence of nodal involvement might be enough for LC staging independently by hilar or mediastinal involvement, and the adoption of scores, which permit prognosis stratification based on different lymph node parameter, may be useful for post-operative management in these patients. We hope that an increased detection rate and LC incidence may permit verification of this hypothesis resulting in a specific staging system for these tumours.

In our analysis, the OS score did not reach statistical significance, even if a clear curve separation and a HR of 1.9 was demonstrated comparing the LRG with the HRG (Figure 2). The low mortality rate of 23 (8.1%) patients in this dataset compared with the previous study (31 cases, 10.5%) [9] may explain the absence of significance, but some considerations might be taken into account. First, the mortality risk is twice as high in the HRG confirming a clear outcome difference between the two groups.

Secondly, in the previous dataset [9] 8 more patients died and 7 of them died of the disease suggesting that increasing the follow-up time mortality will increase. Another consideration regards LC prognosis, which is usually characterized by an indolent evolution that may require long follow-up times also after recurrence; therefore, considering recurrence numbers, it is reasonable to assume that increasing follow-up time, curve separation will increase and more death will occur, which will raise statistical significance. For these reasons, after follow-up continuation statistical significance might be achieved, confirming the curves trend and the validity of the score also for OS. Our results are based on a follow-up of 50 months, which confirms a low disease evolution and the need of a long follow-up time to control LC tumours.

Regarding this point, we can consider the DFS as an appropriate surrogate end-point for LC, taking in account that for the indolent nature of these tumours the OS results may require long follow-up periods to obtain significant results even considering the metastatic disease, with not negligible bias due to non tumour-related deaths [24]. Based on the strong association between DFS and OS resulted in previous studies on neuroendocrine tumours and lung cancers [25,26], the DFS seems to be an effective endpoint in LC, and may be consider extremely reliable for prognosis prediction in these tumours.

This study presents some limitations due to its retrospective nature that did not permit the preoperative management for every patient. To reduce this bias, we screened more than 400 patients limiting the analysis on patients with reported preoperative management and complete data on lymphadenectomy and follow-up. Another limitation regards the extent of lymphadenectomy, sampling and mediastinal node dissection, which is performed on surgeon’s choice because specific lymphadenectomy guidelines for LC are still missing; in consequence, lymph node assessment was based on preoperative clinical staging and surgeons’ experience.

## 5. Conclusions

In this study we validated a prognostic score for LC including patient and tumor characteristics. In particular, in this score a nodal parameter was present in the shape of the node ratio, which may represent a valid option for considering the nodal spread in these tumours. The adoption of composite scores for LC seems to be an effective tool for prognosis prediction in these patients, and may help LC management in different ways, contributing to a global approach to the disease. Indeed, based on our results, future tailored follow-up schedules and post-operative treatments may be planned giving consideration to patient and tumour characteristics. The identification of high-risk patients may permit planning of rigid follow-up schedules and adjuvant treatments that may improve survival rate, while for low-risk patients follow-up may include less testing, reducing costs as well as patient anxiety. However, further prospective studies are needed to validate our findings.

## Figures and Tables

**Figure 1 cancers-14-02601-f001:**
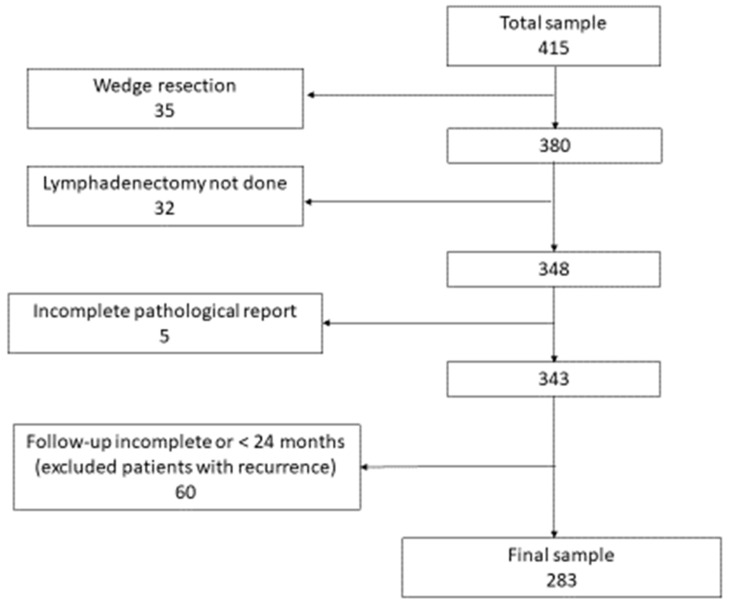
Flow chart showing patient selection for the study.

**Figure 2 cancers-14-02601-f002:**
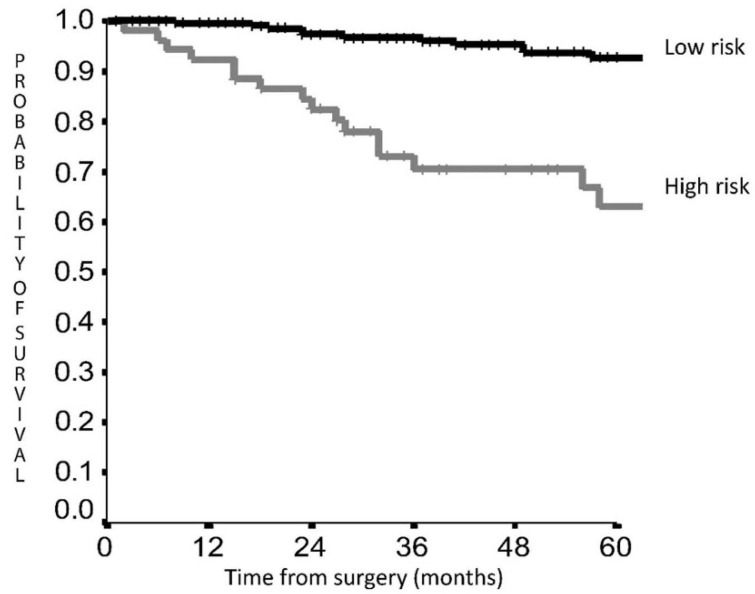
Disease-free survival according to score risk classes.

**Figure 3 cancers-14-02601-f003:**
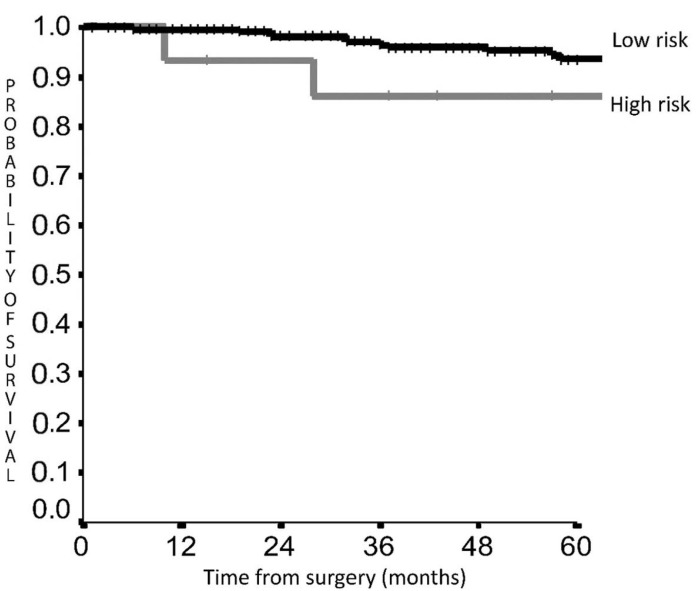
Overall survival according to score risk classes.

**Table 1 cancers-14-02601-t001:** Prognostic scores for overall and disease-free survival.

Overall Survival	Score	Disease-Free Survival	Score
Age > 61 years	1.0	Age > 61 years	1.4
Male Sex	1.0	Atypical Histology	1.0
Lymph Node Ratio > 10%	1.0	Lymph Node Ratio > 10%	1.5
pT stage 2–3	1.4	pT stage 2–3	1.3
**RISK GROUP**		**RISK GROUP**	
LOW RISK	IF SCORE ≤ 3.1	LOW RISK	IF SCORE < 1.5
HIGH RISK	IF SCORE > 3.1	HIGH RISK	IF SCORE ≥ 1.5

**Table 2 cancers-14-02601-t002:** Clinical and pathological characteristics.

Male/Female	116/167
**Age** (years, median)	62 (18–87)
<61 years	135 (47.7%)
**Histology**	
Typical carcinoid	239 (84.5%)
Atypical carcinoid	44 (15.5%)
**Location**	
Central	157 (55.2%)
Peripheral	126 (44.8%)
**Clinical Stage**	
**I**	226 (79.8%)
**II**	41 (14.9%)
**III**	15 (5.3%)
**Surgery**	
Segmentectomy	27 (9.5%)
Lobectomy	238 (84.1%)
Bilobectomy	13 (4.6%)
Pneumonectomy	5 (1.8%)
**Pathological T stage**	
T1	227 (80.2%)
T2	43 (15.2%)
T3–4	13 (4.6%)
**Pathological stage**	
I	235 (83%)
II	30 (10.6%)
III	18 (6.4%)
**NODAL CHARACTERISTICS**	
**pN**	
0	245 (86.6%)
1	23 (8.1%)
2	15 (5.3%)
**N resected nodes**	
<10	160 (56.5%)
≥10	123 (43.5%)
**N positive nodes**	
0	245 (86.6%)
1	21 (7.4%)
>1	17 (6%)
**N resected stations**	
≤3	90 (31.8%)
>3	193 (68.2%)
**N positive stations**	
0	245 (86.6%)
1	28 (9.9%)
>1	10 (3.5%)
**Node ratio**	
<10%	251 (88.7%)
>10%	32 (11.3%)
**Kind of lymphadenectomy**	
Radical node dissection	217 (76.7%)
Sampling/lobe specific	66 (23.3%)

**Table 3 cancers-14-02601-t003:** Patient numbers according to risk categories: LRG: low risk group; HRG: high risk group.

Overall Survival			Disease-Free Survival		
Variable	# Patients (%)		Variable	# Patients (%)	
	**LRG (#268)**	**HRG (#15)**		**LRG (#230)**	**HRG (#53)**
Age			Age		
>61 years	126 (47.3%)	2 (8.3%)	>61 years	96 (41.8%)	41 (78%)
<61 years	142 (52.7%)	13 (91.7%)	<61 year	134 (58.2%)	12 (22%)
Sex			Histology		
Male	101 (37.9%)	15 (100%)	Typical	212 (91.8%)	32 (60%)
Female	167 (62.1%)	0 (0%)	Atypical	18 (8.2%)	21 (40%)
Lymph Node Ratio			Lymph Node Ratio		
>10%	27 (10.1%)	11 (72.7%)	>10%	7 (3%)	6 (12%)
<10%	241 (89.9%)	4 (27.3%)	<10%	223 (97%)	47 (88%)
pT stage			pT stage		
1	82.2%	0 (0%)	1	209 (91%)	38 (72%)
2–3	17.8%	15 (100%)	2–3	21 (9%)	15 (28%)

## Data Availability

Data are property of the participant institutions and may be consulted if required, after authors’ approval.

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
