# Peer review of "External Validation of a Prognostic Score for Survival in Lung Carcinoids"

_cancers, 2022, doi:10.3390/cancers14112601_

Round 1

Reviewer 1 Report

The submitted manuscript entitlled "External validation of a prognostic score for survival in lung carcinoids" is the clinical study of Lung carcinoids. Although authors have tried to show the lung carcinoids data since 2005 to 2019. But their management is not up to the mark. Authors need to add the detailed study in table form which can clearly show the date of collection of samples, patients detail, their age, histology, node ratio and pT for disease-free survival (DFS), age, sex, node ratio and pT for overall survival (OS) etc. it’s better to shows more and possible detail for to well understand the readers. 

In my opinion, the submitted manuscript needs major changes and add more and valuable data’s for the publications to this Journal otherwise they can send this manuscript to the more suited clinical Journals.  

Below are some suggestions for the modifications of the manuscript.

  1. Introduction is not impassive, also doesn’t show the objective of the work. So it must need to write in more details.
  2. Materials and Methods is week and it needs to add more information’s for better understanding of readers.
  3. The studied parameters and their observations need to show in detail.
  4. A detailed descriptions are required for clinical study of the lung carcinoids.
  5. Provided Table 1 and Figure 1 are not enough to understand the detailed things.
  6. Figure 2 and 3, Y-axis are need to edit well.
  7. Conclusion is not up to the mark. Authors need to elaborate and give, current experimental observations of this study and what will be the future prospective of this work. How this study is valuable for the scientific community.
  8. There is no detailed statistical calculation/parameters study in this manuscript.
  9. English need to recheck for the whole manuscript.    

Author Response

REVIEWER 1

The submitted manuscript entitled "External validation of a prognostic score for survival in lung carcinoids" is the clinical study of Lung carcinoids. Although authors have tried to show the lung carcinoids data since 2005 to 2019. But their management is not up to the mark. Authors need to add the detailed study in table form which can clearly show the date of collection of samples, patients detail, their age, histology, node ratio and pT for disease-free survival (DFS), age, sex, node ratio and pT for overall survival (OS) etc. it’s better to shows more and possible detail for to well understand the readers. 

In my opinion, the submitted manuscript needs major changes and add more and valuable data’s for the publications to this Journal otherwise they can send this manuscript to the more suited clinical Journals.  

Below are some suggestions for the modifications of the manuscript.

Comment 1: Introduction is not impassive, also doesn’t show the objective of the work. So it must need to write in more details.

Answer 1: Thank you for your suggestion. We revised the introduction with a better explanation of the background and the importance of the study.

Changes 1: yes introduction

Comment 2: Materials and Methods is week and it needs to add more information’s for better understanding of readers.

Answer 2: thank you for your suggestion, we implemented methods section and  reported how the score was built.

Changes 2: yes

Comment 3: The studied parameters and their observations need to show in detail.

Answer 3: thank you for your comment, we reviewed the results section and the tables implementing the information.

Changes 3: yes table 2

Comment 4: A detailed descriptions are required for clinical study of the lung carcinoids.

Answer 4: thank you, we implemented methods and results sections.

Changes 4: yes table

Comment 5: Provided Table 1 and Figure 1 are not enough to understand the detailed things.

Answer 5: we implemented both.

Changes 5: yes.

Comment 6: Figure 2 and 3, Y-axis are need to edit well.

Answer 6: we rebuilt the figure starting the curves from the beginning. How the Y axis should be improved?

Changes 6 :  yes

Comment 7: Conclusion is not up to the mark. Authors need to elaborate and give, current experimental observations of this study and what will be the future prospective of this work. How this study is valuable for the scientific community.

Answer 7: thank you for your comment, we changed the conclusions.

Changes 7: yes conclusion

Comment 8: There is no detailed statistical calculation/parameters study in this manuscript.

Answer 8: we implemented the statistical section reporting how the model was built and in the results the patient characteristic in every group.

Changes 8: yes, methods

Comment 9: English need to recheck for the whole manuscript.   

Answer 9: thank you, we revised the language.

Changes 9: yes

Reviewer 2 Report

This study confirmed the potential validity of the previous proposed prognostic score for LC by means of an external dataset, for survival prediction in LC patients. However, this study only verified the the previous proposed prognostic score for LC, and did not further optimize or improve, it appears that the depth and innovation of the study is insufficient.

Author Response

REVIEWER 2

Comment 1: This study confirmed the potential validity of the previous proposed prognostic score for LC by means of an external dataset, for survival prediction in LC patients. However, this study only verified the the previous proposed prognostic score for LC, and did not further optimize or improve, it appears that the depth and innovation of the study is insufficient.

Answer 1: Thank you for your comment and for the possibility to improve the manuscript. We tried, on the base of your comment, to improve the manuscript and better describe how the use of this score may improve the lung carcinoid management. In particular, we implemented the introduction with the aim to better explain the importance of these scores. Indeed, despite lung carcinoids present a different clinical behavior and prognosis compared to NSCLC, they are staged using the TNM for LC, that presents important limitations due to sub-stages curves overlapping for these lung carcinoids. Moreover, it seems clear that the nodal parameter is fundamental in these tumors, but it presents a prognostic value independently from the anatomical classification (N1 or N2 according IASLC map). For these reasons, composite prognostic scores were proposed for a better prognosis definition for these patients. In particular, this score presented the innovation of using a different nodal parameter instead the anatomical classification, the node ratio, that seems to be a reliable parameter also for other tumors. Moreover, the identification of high risk patients may be extremely useful to plan tailored follow-up schedules and adjuvant therapy. Indeed, especially regarding the last point, not defined results are present regarding the adjuvant therapy administration, and one of the most interesting points regards the patients selection, that it is still unclear. The use of prognostic scores, and in particular one with different nodal parameters examination, may be useful for an appropriate patients selection for adjuvant therapies. We think that for this reason an external validation of this score was important, to point the bases for possible future studies.

Changes 1: yes, introduction and discussion.

Reviewer 3 Report

Authors utilized the published method of calculating prognostic score for DFS and OS in lung carcinoids. I am not sure how much this paper is relevant to the audience of Cancers journal. 

few comments:

  1. Results are encouraging, authors may want to discuss clinical application of this scoring and how can this approach improve the prognosis and treatment options?
  2. Did authors or other group tried to see if this method have similar results in other subtype of cancer? In other words, any internal control to prove that this method is relevant only for LC, but not for others subtypes?
  3. How does treatment options could be better understood by using this score?
  4. Hard to understand how they calculated score each patient, should provide complete data in Supplementary information.

Author Response

REVIEWER 3

Authors utilized the published method of calculating prognostic score for DFS and OS in lung carcinoids. I am not sure how much this paper is relevant to the audience of Cancers journal. 

few comments:

comment 1:

Results are encouraging, authors may want to discuss clinical application of this scoring and how can this approach improve the prognosis and treatment options?

Answer 1: thank you for your comment. Absolutely, we implemented the discuss section focusing on the treatment options improvement. In particular, we underline the importance of the identification of a high risk group for specific and more close follow-up schedules and especially for the possible identification of high risk patients for adjuvant therapy administration

Changes1: yes, discussion

Comment 2: Did authors or other group tried to see if this method have similar results in other subtype of cancer? In other words, any internal control to prove that this method is relevant only for LC, but not for others subtypes?

Answer 2: thank you for this question. Yes, we performed the inverse thinking regarding this score. Indeed our group previously presents a prognostic scores for NSCLC, identifying different subgroups of patients and different scores that may permit to better stratify prognosis in these patients.

We thought that if these scores were valid for NSCLC, a specific score should work also for lung carcinoids, even if they present a different clinical behaviour. Moreover, we think that the node ratio is a very important factor especially in lung cancer, and we proposed different studies to demonstrate our hypothesis. As you can understand, we investigated for every histology and stage the possible variables that might be included for their prognostic value.

These are our previously published studies:

  1. Pilotto S, Sperduti I, Novello S, Peretti U, Milella M, Facciolo F, Vari S, Leuzzi G, Vavalà T, Marchetti A, Mucilli F, Crinò L, Puma F, Kinspergher S, Santo A, Carbognin L, Brunelli M, Chilosi M, Scarpa A, Tortora G, Bria E. Risk Stratification Model for Resected Squamous-Cell Lung Cancer Patients According to Clinical and Pathological Factors. J Thorac Oncol. 2015 Sep;10(9):1341-1348. doi: 10.1097/JTO.0000000000000628. PMID: 26200453.
  2. Pilotto S, Sperduti I, Leuzzi G, Chiappetta M, Mucilli F, Ratto GB, et al. Prognostic Model for Resected Squamous Cell Lung Cancer: External Multicenter Validation and Propensity Score Analysis exploring the Impact of Adjuvant and Neoadjuvant Treatment. J Thorac Oncol. 2018 Apr;13(4):568-575. doi: 10.1016/j.jtho.2017.12.003. Epub 2017 Dec 18. PMID: 29269009.
  3. Bria E, Milella M, Sperduti I, Alessandrini G, Visca P, Corzani F, Giannarelli D, Cerasoli V, Cuppone F, Cecere FL, Marchetti A, Sacco R, Mucilli F, Malatesta S, Guetti L, Vitale L, Ceribelli A, Rinaldi M, Terzoli E, Cognetti F, Facciolo F. A novel clinical prognostic score incorporating the number of resected lymph-nodes to predict recurrence and survival in non-small-cell lung cancer. Lung Cancer. 2009 Dec;66(3):365-71. doi: 10.1016/j.lungcan.2009.02.024. Epub 2009 Mar 27. PMID: 19327866.
  4. Chiappetta M, Leuzzi G, Sperduti I, Bria E, Mucilli F, Ratto G, Lococo F, Filosso P, Spaggiari L, Facciolo F. Validation of a prognostic model including the number of harvested lymph-nodes in the setting of non-small cell lung cancer patients undergoing curative resection: a multicentre analysis. Minerva Surg. 2021 Aug 2. doi: 10.23736/S2724-5691.21.08902-4. Epub ahead of print. PMID: 34338459.
  5. Chiappetta M, Leuzzi G, Sperduti I, Bria E, Mucilli F, Lococo F, Spaggiari L, Ratto GB, Filosso PL, Facciolo F. Lymph-node ratio predicts survival among the different stages of non-small-cell lung cancer: a multicentre analysis†. Eur J Cardiothorac Surg. 2019 Mar 1;55(3):405-412. doi: 10.1093/ejcts/ezy311. PMID: 30202953.
  6. Chiappetta M, Lococo F, Leuzzi G, Sperduti I, Petracca-Ciavarella L, Bria E, Mucilli F, Filosso PL, Ratto GB, Spaggiari L, Facciolo F, Margaritora S. External validation of the N descriptor in the proposed tumour-node-metastasis subclassification for lung cancer: the crucial role of histological type, number of resected nodes and adjuvant therapy. Eur J Cardiothorac Surg. 2020 Dec 1;58(6):1236-1244. doi: 10.1093/ejcts/ezaa215. PMID: 32770184.
  7. Chiappetta M, Lococo F, Leuzzi G, Sperduti I, Bria E, Petracca Ciavarella L, Mucilli F, Filosso PL, Ratto G, Spaggiari L, Facciolo F, Margaritora S. Survival Analysis in Single N2 Station Lung Adenocarcinoma: The Prognostic Role of Involved Lymph Nodes and Adjuvant Therapy. Cancers (Basel). 2021 Mar 16;13(6):1326. doi: 10.3390/cancers13061326. PMID: 33809513; PMCID: PMC7998125.

Changes 2: no

Comment 3: How does treatment options could be better understood by using this score?

Answer 3: thank you for your question. It is well known that the patients candidates for adjuvant therapy are still undefined. As ESMO and other guidelines reported, the actual indication for adjuvant therapy regards patients at high recurrence risk. For this reason we think that this score, actually validated, permits to identify patients at high recurrence risk and may be selected for adjuvant therapy. We added this part in the discussion.

Changes 3: yes, discussion

Comment 4: Hard to understand how they calculated score each patient, should provide complete data in Supplementary information

Changes 4: thank you for your suggestion. We implemented the statistical section reporting in detail how the score was generate. The score was calculated making the sum of the different prognostic factors reported in table 1.

Answer 4: yes, methods.

Round 2

Reviewer 1 Report

The changes made by authors are fine and i recommend for the publication of this manuscript. 

Reviewer 2 Report

Accept in present form